# The *rsmA* mutant from *Pseudomonas aeruginosa* ID4365 is a non-virulent strain that is suitable for pyocyanin and phenazine-1-carboxylic acid production

Misael Josafat Fabian Del-Olmo[1], Luis Fernando Montelongo-Martínez[1], Gisela Viridiana Villafañe-Barajas[1], Abigail González-Valdez[2], Veronica Roxana Flores-Vega[3,4], Roberto Rosales-Reyes[3], Marco Elias Gudiño-Zayas[3], Thelma Rizo-Pica[5], Raunel Tinoco-Valencia[6], Adelfo Escalante[6], Gloria Soberón-Chávez[2], Miguel Cocotl-Yañez[1]*

1 Departamento de Microbiología y Parasitología, Facultad de Medicina, Universidad Nacional Autónoma de México, Ciudad de México, México, 2 Departamento de Biología Molecular y Biotecnología, Instituto de Investigaciones Biomédicas, Universidad Nacional Autónoma de México, Ciudad de México, México, 3 Unidad de Investigación en Medicina Experimental, Facultad de Medicina, Universidad Nacional Autónoma de México, Ciudad de México, México, 4 Departamento de Biomedicina Molecular, Centro de Investigación y de Estudios Avanzados del IPN, Ciudad de México, México, 5 Hospital General de México Dr. Gerardo Lizeaga, Ciudad de México, México, 6 Departamento de Ingeniería Celular y Biocatálisis, Instituto de Biotecnología, Universidad Nacional Autónoma de México, Cuernavaca, Morelos, Mexico

* mcocotl@comunidad.unam.mx

## Abstract

Phenazines are compounds of medical and biotechnological interest due to their antimicrobial and antitumoral properties. *Pseudomonas aeruginosa*, a Gram-negative bacterium, naturally produces phenazines such as pyocyanin and phenazine-1-carboxylic acid (PCA). These phenazines can generate reactive oxygen species, which increase oxidative stress in susceptible cells and ultimately lead to their death. Pyocyanin can inhibit the growth of human bacterial pathogens, while PCA serves as a potent antifungal agent that can prevent and control crop diseases. The environmental strain *P. aeruginosa* ID4365 overproduces pyocyanin, and its derivative strain IDrsmA, which carries a mutation in the *rsmA* gene, presents a fivefold increase in pyocyanin production compared to the wild-type strain. This mutant is therefore a promising candidate strain for enhancing phenazine production. In this work, we found that the IDrsmA strain has an inactive type three secretion system and is non-cytotoxic. Unlike the wild-type strain ID4365, the IDrsmA strain was unable to infect and kill *Galleria mellonella* larvae or mice. Besides pyocyanin, the IDrsmA strain also produced increased levels of PCA and 1-hydroxyphenazine. In addition, we showed that the deficient phosphate medium, PPGAS, is the best phenazine production medium compared to PPM or King A media. By optimizing culture conditions, we increased pyocyanin production up to 298 μg/mL using strain IDrsmA. We further modified the IDrsmA strain to produce only PCA, achieving a production level of 303

**Data availability statement:** All relevant data are within the paper and its Supporting information files.

**Funding:** MC-Y research was supported by Programa de Apoyo a Proyectos de Investigación e Innovación Tecnológica (PAPIIT) DGAPA, Universidad Nacional Autónoma de México (UNAM), grant number IA200823. GS-Ch research was supported in part by DGAPA, PAPIIT UNAM grant IN202325 and by SECIHTI grant CBF-2025-I28. The funders had no role in the study design, data collection and analysis, decision to publish, or preparation of the manuscript.

**Competing interests:** The authors have declared that no competing interests exist.

µg/mL, which is 10 times higher than that of the wild-type strain. Thus, the IDrsmA strain is non-virulent and suitable for phenazine production.

## Introduction

Phenazines are compounds produced by bacteria and archaea with heterocyclic rings containing nitrogen. These molecules exhibit distinct physical and chemical properties, which depend on their functional groups and include redox potential, acid-base equilibrium, and coloration, among others [1,2]. *Pseudomonas aeruginosa*, a Gram-negative bacterium, is a natural producer of phenazines but is also an opportunistic pathogen in humans. This bacterium produces several virulence factors, including phenazines and the type three secretion system (T3SS), that are important to establish infections [3,4].

Pyocyanin is the primary phenazine produced by *P. aeruginosa* and is exclusive of this species [5]. This blue pigment exhibits redox activity, and when it interacts with molecular oxygen, it produces reactive oxygen species (ROS). These increase oxidative stress in both eukaryotic and prokaryotic cells exposed to this compound, ultimately leading to cell death [6,7]. Due to its redox property, it has been demonstrated that pyocyanin can inhibit the growth of human bacterial pathogens, such as *Staphylococcus aureus*, *Klebsiella pneumoniae*, and *Candida albicans*, as well as the growth of the rice blast fungus, *Magnaporthe griseae* [8,9]. One of the most promising applications of pyocyanin is its use as a safe aquaculture drug [10]. Additionally, pyocyanin can inhibit tumor cell proliferation and induce apoptosis in tumor cell lines, indicating its potential use as an antitumoral agent [11]. Furthermore, this phenazine can be used as a bio-dye, biosensor, and as a component of microbial fuel cells [9,12]. Therefore, pyocyanin presents a wide range of potential applications across diverse biotechnological and medical fields. For *P. aeruginosa* cells, this compound plays a role in biofilm formation and acts as a signal molecule that activates a set of genes in response to oxidative stress. It also functions as an electron acceptor in low-oxygen conditions, among other roles [13–15].

In addition to pyocyanin, *P. aeruginosa* can produce other four phenazines: phenazine-1-carboxamide (PCN), 1-hydroxyphenazine (OH-PHZ), 5-methylphenazine-1-carboxylic acid betain (5-MPCA), and phenazine-1-carboxylic acid (PCA) [5]. PCA is the precursor for all phenazines synthesized by this bacterium and is an effective biopesticide that can help prevent and control crop diseases. In China, this compound, which is commercialized under the name Shenzimycin, was registered and certified as a biopesticide by the Ministry of Agriculture in 2011 [16]. Besides *P. aeruginosa*, PCA is also produced by other species of *Pseudomonas*, including *P. chlororaphis* and *P. fluorescens* [5].

For phenazine synthesis, *P. aeruginosa* has two redundant operons, *phazA1-G1* (*phz1*) and *phzA2-G2* (*phz2*), that encode for the enzymes that convert chorismic acid into PCA. In turn, PCA serves as the substrate for the enzyme PhzM, which produces 5-MPCA, and PhzS further converts this phenazine to pyocyanin [5]. Phenazine synthesis is tightly regulated at the transcriptional level by the three

quorum-sensing systems, Las, Rhl, and Pqs, which control the transcription of both *phz* operons [17,18]. Moreover, the Rsm system also regulates phenazine production at the post-transcriptional level [19,20].

The Rsm system is comprised of four non-coding RNAs (RsmV, RsmW, RsmY, and RsmZ) and two RNA-binding proteins (RsmA and RsmN). RsmA is the primary effector of this system, controlling several virulence traits in *P. aeruginosa*, including biofilm formation, the T3SS, rhamnolipids, and lipase synthesis [21–23]. Additionally, RsmA negatively regulates phenazine production in both the PAO1 and M18 strains. In the first case, *rsmA* inactivation increases the expression of the autoinducers synthases for the Las and Rhl quorum-sensing systems, resulting in higher pyocyanin production compared to the wild-type strain [21]. In the environmental strain *P. aeruginosa* M18, RsmA directly activates the expression of the *phz2* operon but negatively regulates the *phz1* operon. However, PCA is highly produced in the *rsmA* mutant strain [19].

Due to the potential applications of PCA and pyocyanin in medical and biotechnological fields, various strategies have been implemented to enhance their production using non-virulent strains. These strategies include optimizing culture conditions, such as temperature and incubation time, as well as varying carbon sources [2,24,25]. Also, natural producer strains can be genetically engineered to increase phenazine production. In this context, *P. aeruginosa* ID4365 is a marine strain that naturally overproduces pyocyanin in LB medium compared to the PAO and PA14 strains [26,27]. This environmental strain is a natural *lasR* mutant [27] and belongs to phylogroup two, which carries the *exoU* gene encoding the ExoU exotoxin, secreted through the T3SS [28].

It has been reported that when *rsmA* is inactivated in the ID4365 strain, pyocyanin production increases fivefold compared to the wild-type strain. This effect is particularly strong when the strains are grown in the phosphate-deficient PPGAS medium [20]. In addition, *rsmA* inactivation reduces the production of virulence factors, such as elastase and rhamnolipids, due to decreased levels of the Rhl system regulator, RhlR [27]. In this study, we demonstrate that the *rsmA* mutation abolishes ExoU production. This results in significantly reduced cytotoxicity and virulence, making it a non-virulent strain. Furthermore, by optimizing the culture conditions and inactivating genes involved in phenazine synthesis, we increased the production of pyocyanin and PCA. Thus, our results enable us to propose that this mutant strain, named IDrsmA, is a safe strain for overproducing phenazines.

## Materials and methods

### Bacterial strains and growth conditions

Bacterial strains were routinely pre-cultured in Luria-Bertani (LB) medium at 37°C with orbital shaking for 16 hours. When required, antibiotics were added to the LB medium as follows: for *Escherichia coli*, 50 µg/mL apramycin (Ap), 30 µg/mL gentamicin (Gm), and 200 µg/mL carbenicillin (Cb); for *Pseudomonas aeruginosa*, 200 µg/mL Ap, 100 µg/mL Gm, 200 µg/mL Cb, and 120 µg/mL tetracycline (Tc). For phenazine quantification assays, pre-cultured cells were washed and used to inoculate 125 mL flasks containing 30 mL of PPGAS medium ($NH_4Cl$ 20 mM, KCl 20 mM, Tris-HCl pH 7.2 120 mM, $MgSO_4$ 1.6 mM, glucose 0.5%, and peptone 1.0%), PPM medium (triptone 22 g/L, glucose 20 g/L, $KNO_3$ 5 g/L pH 7.5) or King A medium (peptone 20 g/L, $K_2SO_4$ 10 g/L, $MgCl_2$ 1.64 g/L pH 7.3) each with an initial $OD_{600}$ = 0.05. Bacterial strains and plasmids are listed in S1 and S2 Tables, respectively.

### DNA manipulation techniques

Standard molecular biology protocols were used for plasmid and genomic DNA manipulation. Plasmid DNA was isolated using the Wizard® Plus SV Minipreps DNA Purification System (Promega), while genomic DNA was isolated using the GeneJET Genomic DNA Purification Kit (Thermo Scientific). Restriction enzymes (New England Biolabs) and T4 DNA ligase (Promega) were used according to the manufacturer's instructions. PCR amplification of cloning fragments was carried out with Phusion Hot Start II DNA Polymerase (Thermo Scientific), and GoTaq® Flexi DNA Polymerase (Promega) was employed for colony PCRs. All plasmids were propagated in *E. coli* DH5α and introduced into *P. aeruginosa* by

electroporation. Oligonucleotides synthesis and DNA sequencing were performed at the Unidad de Síntesis y Secuenciación de DNA (USSDNA) by the Instituto de Biotecnología, Universidad Nacional Autónoma de México (UNAM). Oligonucleotides are listed in the S3 Table.

## Western blot assay

Experiments were conducted as previously described [29], with minor modifications. Briefly, *P. aeruginosa* strains were cultured in LB broth supplemented with 5 mM EGTA and 20 mM MgCl$_2$ (induction medium) at 37 °C with shaking at 225 rpm. At the early stationary phase (OD$_{600nm}$≈1), 1.0 mL aliquots centrifuged at 14,000 rpm for 2 min at 4 °C. Pellets were washed twice with sterile induction medium, dried, and stored at –20 °C. Supernatants were carefully transferred to sterile tubes and centrifuged again to remove residual cells. Secreted proteins were precipitated with 10% (v/v) trichloroacetic acid (TCA) at 4 °C for 16 hours and then centrifugated at 14,000 rpm for 30 min at 4 °C. Pellets and protein precipitates were resuspended in loading buffer, with volume adjusted according to the optical density at 600 nm of to ensure equal loading. Residual TCA from secreted protein samples was neutralized by adding 10% (v/v) saturated Tris solution. Samples were heated at 92 °C for 5 min before being loaded onto 12% SDS-polyacrylamide gels, as per Bio-Rad protocols. Proteins were transferred onto 0.2 μm nitrocellulose membranes and blocked with 5% bovine serum albumin in TBS-T buffer (20 mM Tris-HCl, 150 mM NaCl, pH 7.4, 0.1% Tween-20) at 4 °C for 2 hours. Membranes were incubated with anti-ExoU polyclonal antibody (1:10,000 dilution) and anti-OprF antibody (1:5,000 dilution; MyBioSource) for 1 hour at 4 °C. After washing, membranes were incubated with an AP-conjugated anti-rabbit IgG (1:5,000 dilution; Abcam) at 4 °C for 1 hour. Detection was performed using a 1-Step NBT/BCIP substrate solution (Thermo Scientific).

## Cytotoxicity assay

The cytotoxicity assays were performed as reported previously [29]. Briefly, *P. aeruginosa* strains were grown in LB broth at 180 rpm and 37 ºC, and the bacteria were adjusted to a multiplicity of infection (MOI) of 50 to infect $6 \times 10^5$ HeLa cells (ATCC® CCL-2™) for 2 hours. Supernatants of infected cells were collected and centrifuged at 14,000 rpm for 2 min. Cytosolic lactate dehydrogenase (LDH) activity (Promega) was quantified. The percentage of LDH activity was determined using the following formula: % of LDH release = (OD$_{490}$-experimental LDH release OD$_{490}$-spontaneous LDH release)/ (OD$_{490}$-maximal LDH release - OD$_{490}$-spontaneous LDH release) x 100%.

## Virulence assays

For *Galleria mellonella* infection assays, groups of 10 larvae were inoculated with 10 μL of a bacterial suspension ($1 \times 10^2$ CFU) into the hemocoel using an insulin syringe. The infected larvae were incubated at 37°C, and survival was recorded at 24 and 48 h post-infection. The PA14 and ATCC 9027 strains were used as positive and negative controls, respectively. The assay was performed in duplicate.

For a mouse model of pneumonia. All mouse experiments were conducted in accordance with the ARRIVE guidelines and the standard operating protocols approved by the Comité de Investigación y Ética of the Facultad de Medicina, Universidad Nacional Autónoma de México (UNAM). The use of animals was conducted under a protocol approved by the Comité Interno para el Cuidado y Uso de Animales de Laboratorio (CICUAL). Four groups of five BALB/c randomized mice, housed by cage, were anesthetized with 20 mL of intraperitoneal ketamine (6.7 mg/mL) and xylazine (1.3 mg/mL). On day 0, mice were inoculated with either 20 μL of PBS (control group) or 10 μL of a bacterial suspension ($1 \times 10^{\wedge 7}$ CFUs, 10 μL per nostril). Animals were maintained under controlled temperature conditions to ensure their comfort and minimize stress, and circadian cycles were regulated through controlled light-dark periods. The humane endpoints to cull mice before the end of the six-day infection were: significantly reduced or loss of activity, abnormal breathing, reduced intake (food/water), weight loss (0, 10, and more than 15%), loss of mobility, stooped position, bristly coat, lack of grooming, decreased alertness, dirty orifices (ocular, nasal, and urinary secretions, diarrhea), and cyanotic extremities. Three scores

were given for each of the signs above (0, normal; 1, mild; 2, accentuated). Scores of 0–1 indicated that the mouse was healthy; 2−4 indicated a need for close monitoring every 6 hours; 5−8 showed the need for analgesic treatment (meloxicam 1.5 mg/kg subcutaneously every 12 h). Animals reaching a score of 9 or above were immediately euthanized by exposure to $CO_2$, followed by cervical dislocation. Lungs from all mice were removed and fixed in 5 mL of 4% p-formaldehyde for 24 hours at room temperature. Tissue were embedded in paraffin. Sections (4−6 µm) were stained with hematoxylin and eosin and visualized using a 40X objective on a Nikon Microphot-FXA microscope. Photomicrographs were captured using an MXD1200 camera with ACT-1 Nikon software at a resolution of 3600 × 2880 pixels. Each image, obtained with a digital microscope, covers a peribronchial area of 193.03 × 154.42 µm. Neutrophils presence was digitally analyzed using Fiji ImageJ. Fifteen images from three mice (n = 15) were analyzed.

### Construction of the mutant strain in *phzM*, *phzS*, and *phzH*

To obtain the mutant strain in *phzM*, *phzS*, and *phzH*, three plasmids were constructed to delete each gene as follows: approximately 500 bp upstream and approximately 500 bp downstream of each gene, and an apramycin resistance cassette obtained from the pIJ773 plasmid [30] was amplified by PCR. The three PCR products were purified and used as a template in a nested PCR. The resulting fragment was cloned into pJET1.2 Blunt (Thermo Scientific), generating the plasmids pJetM, pJetS, and pJetH. Then, the plasmids pJetM and pJetH were digested with HindIII to release the mutagenic product and cloned into the HindIII site in the pEX18Amp plasmid [31]. The plasmid pJetS was digested with EcoRI, and the product of interest was cloned into the EcoRI site in the pEX18Sm [32]. The resulting plasmids, pEXM, pEXS, and pEXH, were verified by restriction pattern and sequencing. Next, one µg of the purified pEXM plasmid was electroporated into competent cells of ID4365 and IDrsmA strains, and an apramycin-resistant transformant, generated by a double recombination event, was isolated and confirmed by PCR. Then, the apramycin cassette was removed using the plasmid pFLP2 and verified by PCR. The same strategy was used to delete *phzS* and *phzH*, using the plasmids pEXS and pEXH, respectively, to obtain the strains IDMSH and IDAMSH.

### Quantification of phenazines by HPLC

Phenazines quantification by HPLC was performed as reported previously [33] with minor modifications. Briefly, 1 mL of culture was centrifuged at 14,000 rpm for 4 min, and the supernatant was filtered through 0.22 µm membranes (Cobetter filters, 13 mm). A volume of 500 µL was mixed with 500 µL chloroform, vortexed for 10 seconds, and centrifuged at 14,000 rpm for 8 min. The organic phase was transferred into a new tube and air-dried in a fume hood. This process was repeated 3 times to extract the total of phenazines. Dried samples were stored at −20°C until reconstitution with 300 µL 90% acetonitrile before injection. Chromatographic analyses were carried out on a Varian ProStar 210 system equipped with a UV-Vis detector. Separation was achieved using a C18 reversed-phase column (XBridge® C18 5µm, 4.6x150 mm) with a 20 µL injection volume. The mobile phases consisted of 15% (Solvent A) and 100% (Solvent B) acetonitrile, applied through a linear gradient from 0% to 100% over 15 minutes at a flow rate of 1.5 mL/min. Detection was performed at 365 nm. Phenazine concentrations were calculated from the standard curves from 100 to 1000 µM.

### Statistical analysis

All statistical analyses and corresponding graphs were performed using RStudio (version 4.4.0). Data from three independent biological replicates per strain were evaluated using the Shapiro–Wilk test for normality ($p > 0.05$) and for homogeneity of variances using Levene's test ($p > 0.05$). Results are expressed as the mean ± standard deviation (SD). Depending on data distribution, statistical comparisons between groups were performed using either an unpaired *t*-test, one-way ANOVA, or Kruskal–Wallis' test, with significance set at $p < 0.05$. When ANOVA or Kruskal–Wallis' tests revealed significant differences, post hoc analyses were conducted using Tukey's Honestly Significant Difference (Tukey-HSD) test or Dunn's test, respectively. All raw data generated in this study are available from the authors upon reasonable request.

### Ethics statement

All mouse experiments were conducted in accordance with the ARRIVE guidelines and the standard operating protocols approved by the Comité de Investigación y Ética from the Facultad de Medicina (https://di.facmed.unam.mx/investigacion.php) of the Universidad Nacional Autónoma de México (UNAM), as per official letter No. FMED/CEI/PMSS/140/2022. The use of animals in experimental procedures was conducted under a protocol approved by the Comité Interno para el Cuidado y Uso de Animales de Laboratorio (CICUAL; https://di.facmed.unam.mx/cicual.php), official letter 021-CIC-2022.

## Results

### Inactivation of *rsmA* abolishes ExoU production and cytotoxicity

Previously, it was reported that RsmA is a positive regulator of the T3SS in the clinical strain PAO1 [23]. To determine whether this regulation is conserved in the environmental strain ID4365, we conducted a Western blot under T3SS-induction conditions, using an ExoU antibody to detect its secretion via the T3SS in the strains ID4365, IDrsmA, and their derivatives. We used PA14, the type strain for phylogroup 2 [34], as a positive control and ATCC 9027, which naturally lacks the T3SS [35], as a negative control. As shown in Fig 1, ExoU secretion was abolished in the IDrsmA strain compared to the wild-type strain. This phenotype was restored in the IDrsmA strain when complemented with *rsmA* expressed from the pUCrsmA plasmid but not with the empty plasmid, pUCP20. As expected, the PA14 strain secreted ExoU but not the ATCC 9027 strain. We also performed a Western blot using cell extracts to determine if ExoU was not secreted or not produced in the *rsmA* mutant strain. As shown in S1 Fig, ExoU was detected at very low levels in PA14, ID4365, and IDrsmA/pUCrsmA, indicating nearly complete secretion of the exotoxin. In contrast, ExoU was undetectable in ATCC 9027, IDrsmA and IDrsmA/pUCP20, indicating failure of ExoU synthesis. These results show that, like PAO1, RsmA acts as an activator of the T3SS in the environmental strain ID4365.

ExoU is a phospholipase that triggers cell death in eukaryotic cells and drives T3SS-related cytotoxicity [36]. Since *rsmA* inactivation abolishes ExoU production, we determined whether cytotoxicity was likewise lost in the IDrsmA strain. The same strains from the Western blot assays were incubated with HeLa cells, and lactate dehydrogenase release was measured as a result of cytotoxicity, as described in the Materials and Methods section. Our results showed that, like ATCC 9027, the IDrsmA strain showed no cytotoxicity. This phenotype was restored to ID4365 levels in the IDrsmA strain complemented with pUCrsmA (Fig 1c).

### The IDrsmA is a non-virulent strain

The above data indicate that *rsmA* inactivation may decrease the virulence of the ID4365 strain. To confirm, we performed an *in vivo* virulence assay using the *Galleria mellonella* model. Cells from ID4365 and its derivatives were injected into *G. mellonella* larvae, with PA14 and ATCC 9027 as positive and negative controls, respectively. As shown in Fig 2, after 24 hours, the strains PA14, ID4365, and IDrsmA/pUCrsmA killed all larvae. In contrast, the larvae inoculated with IDrsmA, IDrsmA/pUC20, and ATCC 9027 strains survived up to 48 hours post-infection.

We next used a murine pneumonia model to evaluate the virulence of ID4365, IDrsmA, and their derivatives. Mice were infected by aspiration with $1\times10^7$ CFUs of each strain. At 24 hours post-infection, mice infected with strains IDrsmA and IDrsmA/pUCP20 did not display signs of illness. In contrast, ID4365 and IDrsmA/pUCrsmA caused a 40% lethality rate. Of the remaining mice in these groups, 40% showed illness (not reaching humane endpoints), and 20% remained asymptomatic. Groups infected with IDrsmA and IDrsmA/pUCP20 stayed asymptomatic at 24 hours post-infection. Histological analysis of lung sections showed increased peribronchial neutrophils, vascular congestion, and focal bleeding in all infected groups (Fig 3a). Quantitative analysis showed higher neutrophil counts in mice infected with IDrsmA and IDrsmA/pUCP20 than those infected with ID4365 and IDrsmA/pUCrsmA (Fig 3b).

Overall, our results indicate that the IDrsmA strain is avirulent and can be considered a safe option for phenazine overproduction.

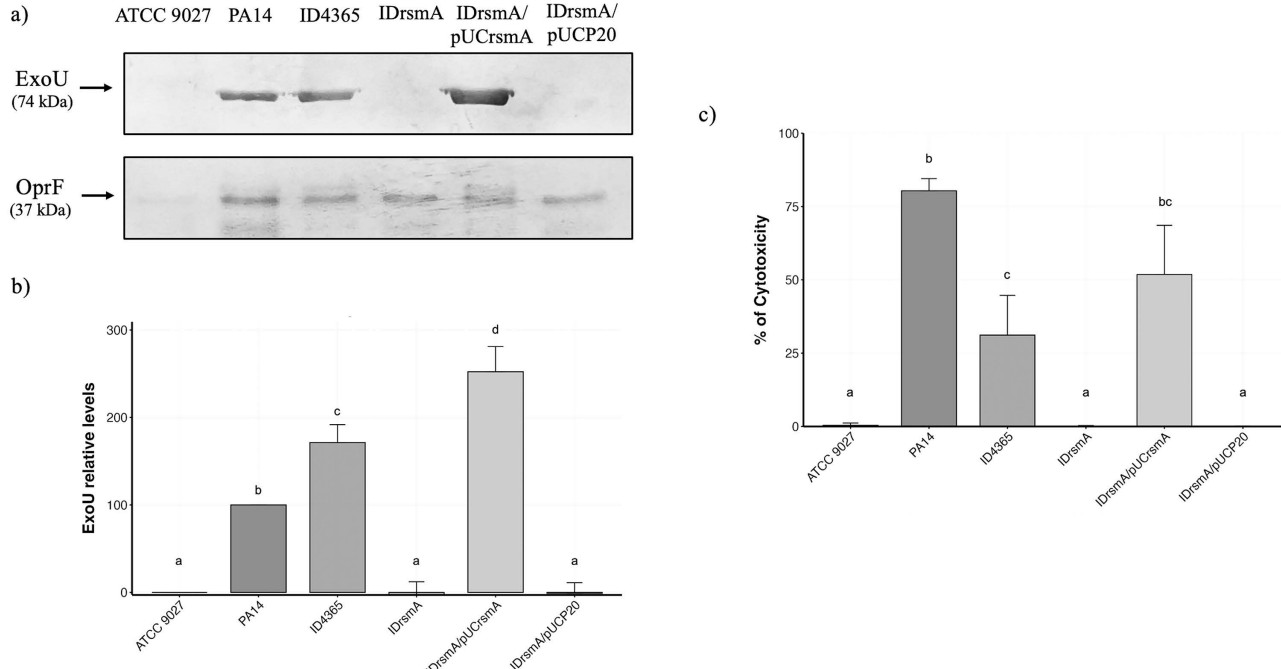

**Fig 1. RsmA inactivation abolishes ExoU secretion and cytotoxicity.** A) Western blot against ExoU from supernatant of strains grown in induction conditions at early stationary phase, OprF was used as a loading control. B) Densitometry analysis of ExoU protein levels normalized against OprF of three independent western blot assays. Bar graphs represent the mean±standard deviation (SD). Statistically significant group differences were determined using one-way ANOVA with the Tukey-HSD post hoc test ($p < 0.05$). Different letters above the bars indicate statistically significant differences, same letters denote no significant difference. C) Cytotoxicity using HeLa cells (ATCC® CCL-2™). Results are presented as the mean±SD of three biological experiments, each performed in duplicates. Statistically significant groups differences were determined with the Kruskal–Wallis test and Dunn´s test ($p < 0.05$). Different letters above the bars indicate statistically significant differences, same letters denote no significant difference.

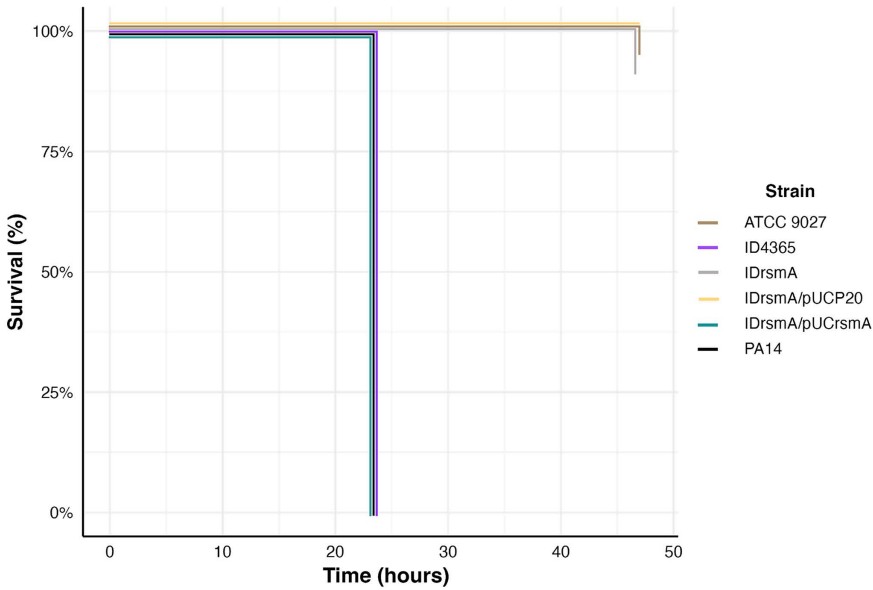

**Fig 2. The IDrsmA strain is non-virulent in *G. mellonella*. *G. mellonella* larvae were infected with 1x10² CFU and their survival was monitored for 24 and 48 hours.** The PA14 and ATCC 9027 strains were used as positive and negative controls of virulence, respectively.

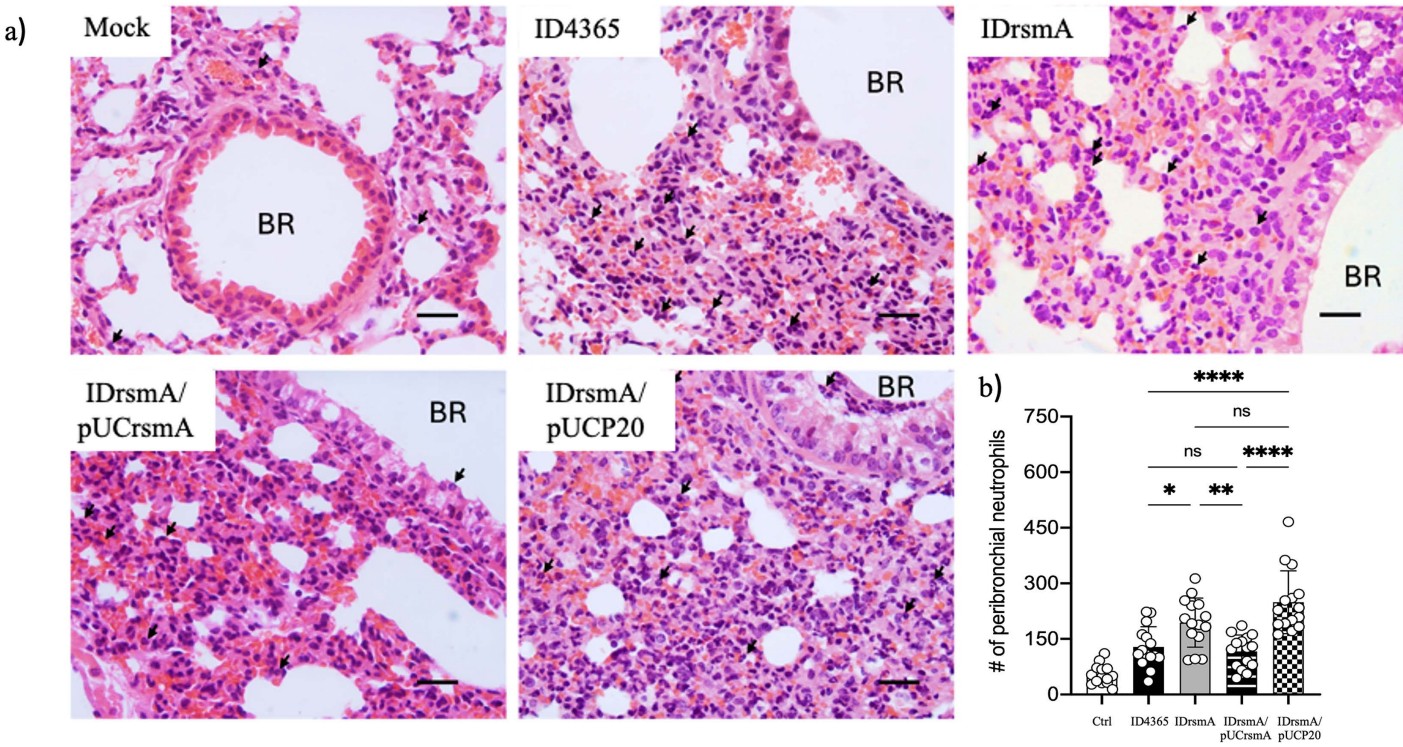

**Fig 3.** *rsmA* **inactivation abolishes virulence in the ID4365 strain.** A) Histological analysis of lung sections of mice infected with *P. aeruginosa* strains. Anesthetized animals were infected with $1\times10^7$ CFUs by the nasal pathway. The left lungs of euthanized animals were processed for histological analysis. Similar lung sections of uninfected (Mock) or mice infected with the ID4365, IDrsmA, and derivatives strains were presented. B) Quantification of neutrophil infiltration is presented. Lungs of three infected mice or controls were processed and stained with H&E. Images were taken with a 40x objective; arrows indicate the presence of neutrophils. Bars indicate 20 μm. BR, bronchiole. The quantification of neutrophils in peribronchial areas was based on five aleatory images from three animals (n = 15). Bar graphs represent the mean ± standard deviation (SD). Statistically significant differences between groups were determined using an unpaired *t*-test (*$p < 0.05$, **$p < 0.01$, ***$p < 0.001$, ****$p < 0.0001$, ns = not significant).

## PPGAS medium, but not PPM or King A medium, increases pyocyanin production in the ID4365 strain

In previous work, we found that pyocyanin production increased when the ID4365 strain was grown in the phosphate-deficient PPGAS medium compared to LB [20]. However, it has been reported that both PPM and King A media promote high phenazine production [9,16]. Therefore, to compare and select the most effective culture medium for phenazine production, we incubated the ID4365 strain in PPGAS, PPM, and King A media for 24 hours at 37 ºC, and pyocyanin production was measured by HPLC as described in the Materials and Methods section. As shown in S2 Fig, pyocyanin is not produced in PPM medium and is produced only slightly in King A compared to the PPGAS medium. Therefore, we chose the PPGAS medium for the subsequent phenazine production assays.

## Inactivation of *rsmA* increases pyocyanin production, as well as OH-PHZ and PCA

Previously, we found that the IDrsmA strain produced fivefold more pyocyanin than the wild-type ID4365 strain [20,27], although this was measured using a method that detects only pyocyanin [37]. To provide a more comprehensive assessment of phenazine production, we used HPLC to measure phenazine production in the IDrsmA strain after 24 hours of growth at 37°C, and we compared the results to those from the wild-type ID4365 strain. Consistent with our earlier findings, we found that pyocyanin production in the IDrsmA strain increased sixfold compared to the parental

strain ID4365. Additionally, inactivation of *rsmA* resulted in a sixfold and eightfold increase in PCA and OH-PHZ production, respectively (Fig 4).

We next extended the incubation time to 72 hours to increase pyocyanin production. As shown in Fig 5, the pyocyanin concentration increased from 189 µg/mL at 24 hours to 298 µg/mL at 72 hours, representing a 58% increase compared to the concentration at 24 hours.

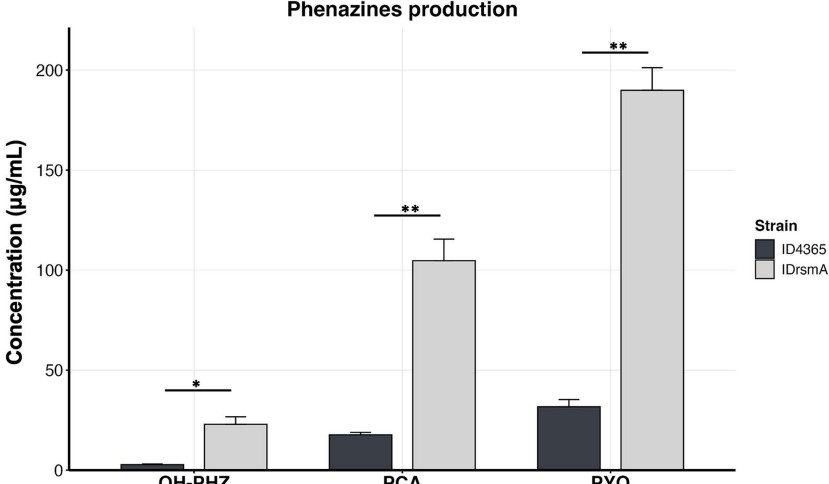

**Fig 4. *rsmA* mutation increases phenazine synthesis in the ID4365 strain.** The strains ID4365 and IDrsmA were grown in PPGAS medium at 37ºC and 225 rpm for 24 hours, and phenazines were extracted using chloroform and analyzed by HPLC. Bar graphs represent the mean ± standard deviation (SD). Statistically significant differences between groups were determined using an unpaired *t*-test (\**p* < 0.05, \*\**p* < 0.01, \*\*\**p* < 0.001, \*\*\*\**p* < 0.0001). OH-PHZ: 1-hydroxyphenazine, PCA: phenazine-1-carboxylic acid, PYO: pyocyanin.

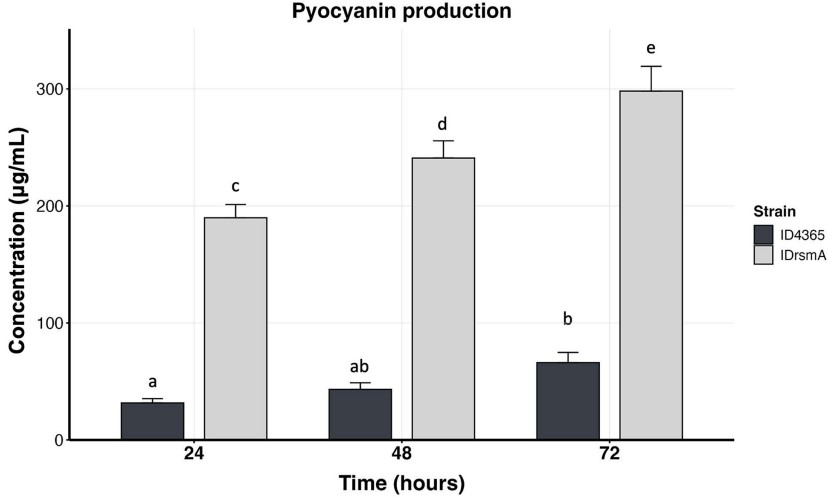

**Fig 5. The IDrsmA strain overproduces pyocyanin in PPGAS medium.** The strains ID4365 and IDrsmA were grown in PPGAS medium at 37ºC and 225 rpm, and pyocyanin was measured by HPLC at 24, 48, and 72 hours. Bar graphs represent the mean ± standard deviation (SD). Statistically significant group differences were determined using one-way ANOVA with the Tukey-HSD post hoc test (*p* < 0.05). Different letters above the bars indicate statistically significant differences, same letters denote no significant difference.

## Improving PCA production in the IDrsmA strain

Since the IDrsmA strain overproduces pyocyanin, we determined whether this strain can also be suitable for PCA production. To abolish the production of pyocyanin, OH-PHZ, and PCN, we sequentially inactivated *phzM*, *phzS*, and *phzH* in the ID4365 and IDrsmA strains, rendering the IDMSH and IDAMSH strains, respectively. We next measured PCA production by HPLC at 24 hours of growth and 37 °C. The chromatogram showed that inactivation of the three genes in both ID4365 and IDrsmA strains allowed the production of only PCA since the peaks corresponding to pyocyanin, OH-PHZ, and PCN were not detected (S3 Fig). In addition, PCA production in the IDMSH strain was twofold compared to the wild-type strain ID4365. Moreover, the IDAMSH strain produced twelvefold PCA compared to the ID4365 strain, twofold compared to the IDrsmA strain, and sixfold compared to the IDMSH strain (Fig 6).

To build upon these findings, we next extended incubation to 72 hours to enhance PCA production. As a result, PCA levels nearly doubled in ID4365, IDMSH, and IDrsmA at 72 hours versus 24 hours. Moreover, the IDAMSH strain increased nearly 45% PCA synthesis at 72 hours compared to 24 hours, reaching a concentration of 303 µg/mL (Fig 6). These findings suggest that the IDAMSH strain is highly suitable for PCA production.

## Discussion

Phenazines are bacterial compounds with redox activity [1]. Pyocyanin, among the most studied phenazines, is synthesized by *P. aeruginosa*, an opportunistic human pathogen [4,5]. Pyocyanin reacts with molecular oxygen and, due to its redox potential, generates ROS, such as hydroxyl radical or superoxide anions. These ROS increase oxidative stress in both eukaryotic and prokaryotic cells exposed to the compound, causing cell death [6]. This characteristic makes pyocyanin a potent antimicrobial agent, capable of inhibiting the growth of several human pathogens, including *S. aureus*, *K. pneumoniae*, *S. typhi*, and *C. albicans*, among others [8,9].

Besides its antimicrobial effects, pyocyanin has potential applications as a biocolorant, antitumor agent, or biosensor [9,11,12]. In addition to pyocyanin, *P. aeruginosa* produces PCA, a broad-spectrum fungicide active against fungi that

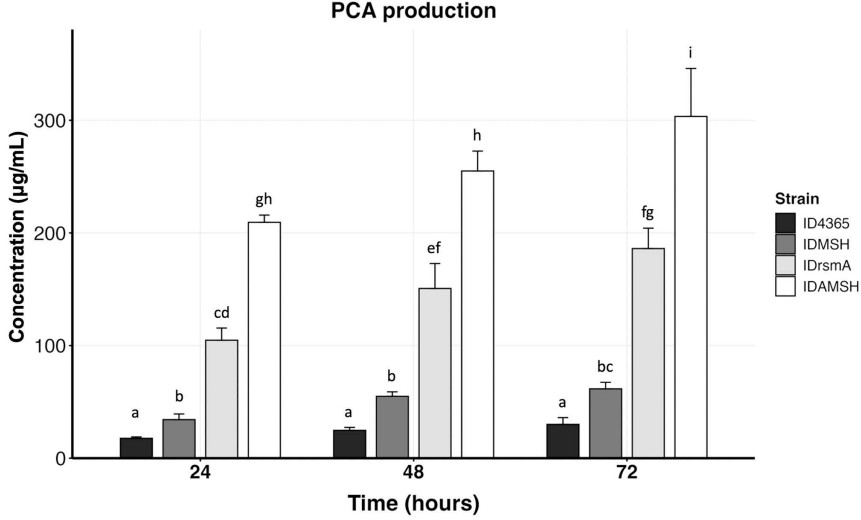

**Fig 6. The IDAMSH strain -an IDrmA derivative strain mutant in *phzM*, *phzS*, and *phzH*- overproduces PCA.** The strains were grown in PPGAS medium at 37 °C and 225 rpm, and PCA was measured by HPLC at 24, 48, and 72 hours. The IDMS strain is derived from the ID4365 strain, which carries deletions in *phzM*, *phzS*, and *phzH*. Bar graphs represent the mean ± standard deviation (SD). Statistically significant group differences were determined using one-way ANOVA with the Tukey-HSD post hoc test ($p < 0.05$). Different letters above the bars indicate statistically significant differences, same letters denote no significant difference.

harm rice and fruit crops [5,16]. Due to these activities, phenazines have potential applications in various fields of bio-technology and medicine. Although phenazines can be chemically synthesized, this process typically yields low amounts and generates toxic byproducts [38]. Thus, genetic modification of producing strains, such as *P. aeruginosa*, is a common strategy.

Previously, we reported that *P. aeruginosa* ID4365, a marine strain and natural mutant in *lasR*, overproduces pyocyanin compared to the clinical strains PAO1 and PA14. When *rsmA* was deleted in the ID4365 strain, resulting in the IDrsmA strain, pyocyanin production increased 5-fold compared to its parental strain and surpassed the production of clinical strains [20,27]. The gene *rsmA* encodes for the post-transcriptional regulator RsmA, a component of the global Rsm system that regulates the synthesis of several virulence factors in *P. aeruginosa* [21,39,40]. In the ID4365 strain, RsmA negatively regulates the translation of the *phz1* operon, as well as the *phzM* and *phzS* genes, all of which are involved in phenazine synthesis. RsmA also positively regulates the production of elastase and rhamnolipids by controlling the levels of the quorum-sensing regulator RhlR [27,29]. Thus, the IDrsmA strain produces low levels of these two virulence factors.

Since the IDrsmA strain overproduces pyocyanin, we determined whether this strain could be suitable for phenazine production. However, the pathogenicity of *P. aeruginosa* represents a challenge for using this bacterium for this purpose. Therefore, we evaluated the pathogenicity of the IDrsmA strain to determine its biotechnological potential. A key virulence factor of *P. aeruginosa* is the secretion of exotoxins through the T3SS [41,42]. Strains defective in this system show reduced pathogenicity [3,43]. Previous studies have shown that RsmA acts as a positive regulator of the T3SS in the clinical strain PAO1 [23]. Thus, we sought to determine whether this regulatory mechanism is conserved in the marine strain ID4365. Our results indicated that *rsmA* inactivation abolishes the production of the cytotoxic ExoU exoenzyme and significantly reduces *in vitro* cytotoxicity. These results suggest that, like the PAO1 strain, RsmA is a positive regulator of the T3SS.

Furthermore, we found that the IDrsmA strain was completely avirulent, as this strain was unable to kill *G. mellonella* larvae and mice. In contrast, the wild-type ID4365 strain was lethal to both organisms. Interestingly, our findings showed that infection with the IDrsmA strain led to increased neutrophil infiltration into the lungs of mice. This suggests that the immune response was heightened, resulting in better control of the *P. aeruginosa* infection. Additionally, these results suggest that, although the IDrsmA strain overproduces pyocyanin, additional virulence factors, such as T3SS, are necessary for this strain to establish an infection. Regarding this, previous studies have shown that mutants lacking functional T3SS machinery exhibit reduced virulence [3,16,43]. Therefore, while we conducted cytotoxicity assays on HeLa cells and used two animal models for virulence testing, it is likely that similar findings could be obtained using other cell or animal models.

The reduced virulence of the IDrsmA strain can be attributed to its low production of the RhlR protein, which in turn reduces elastase and rhamnolipid synthesis, as well as the inactivation of the T3SS, and an inherent *lasR* mutation in the ID4365 strain [20,27]. Together, this makes the IDsmA strain non-virulent and a safe option to overproducing phenazines.

In a previous study, we found that the IDrsmA overproduces pyocyanin in LB medium compared to the wild-type strain, and this production increased even further in phosphate-deficient PPGAS medium [20,27]. Since the culture medium plays a critical role in enhancing phenazine production, we tested two additional media known to be suitable for phenazine production: King A and PPM medium [9,16]. However, our results showed that the ID4365 strain produces a significantly higher amount of pyocyanin specifically in PPGAS medium. Regarding this, previous research has documented that RhlR expression is particularly more active in this medium [44]. Moreover, in low phosphate conditions, pyocyanin production is also increased by the activation of *rhlR* by PhoB, the regulator protein that controls the response to phosphate limitation [45]. Therefore, at least to the marine strain ID4365, the PPGAS medium allows an increase in phenazine concentration.

As stated, the IDrsmA strain overproduces pyocyanin. Furthermore, HPLC results showed that this strain produces more OH-PHZ and PCA than the wild-type ID4365 strain. This suggests that while the inactivation of *rsmA* increases the expression of the *phz1* operon and the *phzM* and *phzS* genes [20], the enzymatic activities of PhzM and PhzS are insufficient to convert all PCA into pyocyanin. Nevertheless, it is important to note that pyocyanin remains the predominant

phenazine produced by the IDrsmA strain. Additionally, we observed an increase in pyocyanin production from 190 μg/mL at 24 hours to 298 μg/mL at 72 hours, representing a 58% increase in production. This level of production is comparable to previously reported pyocyanin-overproducing strains. For example, *Pseudomonas* spp. MCC 3145, which achieved 313.94 ± 10.09 mg/L after statistical media optimization was applied [46].

We were also interested in determining whether the IDrsmA strain can be modified to overproduce PCA. As expected, *phzM*, *phzS*, and *phzH* inactivation abolished pyocyanin synthesis, and PCA was the only phenazine produced in the IDAMSH strain. Furthermore, PCA production at 24 hours by this mutant increased twofold compared to the IDrsmA strain, sixfold compared to the *phzM*, *phzS*, *phzH* mutant strain from ID4365, and nearly twelvefold compared to the wild-type strain ID4365. These findings indicate that *rsmA* inactivation allows PCA accumulation despite the reduced expression of the *phz2* operon in the IDrsmA strain [20]. Furthermore, PCA production increased by approximately 45% at 72 hours compared to 24 hours in the IDAMSH strain, reaching a total of 303 μg/mL. However, this production is lower than that reported by the PA-IV strain, a derivative of *P. aeruginosa* PA1201 carrying several mutations, including the inactivation of the T3SS [16]. Strain PA-IV achieved a production of 9.8 g/L in fed-batch fermentation [16], which represents an optimized production condition and cannot be directly compared to the conditions used in this work.

In summary, our results allow us to propose the IDsmA as a non-virulent strain suitable for producing phenazines. Additional strategies can be implemented to increase pyocyanin and PCA production. Options include overexpressing genes related to the accumulation of chorismic acid, increasing the expression of genes involved in phenazine synthesis, and optimizing fermentation conditions, including the culture medium starting with PPGAS. Finally, the mutant strain derived from IDrsmA, called IDAMSH, which carries mutations in the *phzM*, *phzS*, and *phzH* genes, can serve as a chassis for the overproduction of other phenazines, including those derived from different *Pseudomonas* species.

## Supporting information

**S1 Fig.** *rsmA* **inactivation abolishes ExoU production in the ID4365 strain.** Western blot shows ExoU in the pellet of strains grown under induction conditions at early stationary phase, with OprF used as a loading control. Asterisks indicate the ExoU protein band.
(TIFF)

**S2 Fig. Pyocyanin is mainly produced in PPGAS medium.** Pyocyanin production by the strains ID4365 and IDrsmA was measured in PPM, King A, and PPGAS medium at 37 °C and 24 hours. Statistically differences were determined by one-way ANOVA with Tukey-HSD post hoc test ($p < 0.05$). Different letters above the bars indicate statistically significant differences, same letters denote no significant difference.
(TIFF)

**S3 Fig. The IDAMSH strain produces only PCA.** At 24 hours, its chromatogram shows only PCA, while the IDrsmA strain produces pyocyanin (PYO), PCA and OH-PHZ.
(TIFF)

**S1 Table. Bacterial strains used in this work.**
(DOCX)

**S2 Table. Plasmids used in this work.**
(DOCX)

**S3 Table. Oligonucleotides used in this study.**
(DOCX)

**S1 Raw Images.** PDF file with original, uncropped, and unadjusted images of the Western blot for **Figs 1** and S1 with biological replicates.
(PDF)

**S1 Data.** Excel file with the data set used to generate the graphs of this work.
(XLSX)

## Acknowledgments

MJF-O, LFM-M, and GVV-B are students of the Programa de Maestría y Doctorado en Ciencias Bioquímicas at the Universidad Nacional Autónoma de México (UNAM), and VRF-V is a student of the Departamento de Biomedicina Molecular at CINVESTAV-IPN, all thanks to SECIHTI. The authors would like to thank Dr. Brenda Rosario Sandoval Meza, part of the Medical Translation Area at the Research Division, Faculty of Medicine, Universidad Nacional Autónoma de México, for proofreading this paper.

## Author contributions

**Conceptualization:** Miguel Cocotl-Yanez.

**Data curation:** Luis Fernando Montelongo-Martínez, Miguel Cocotl-Yanez.

**Formal analysis:** Misael Josafat Fabian Del-Olmo, Luis Fernando Montelongo-Martínez, Gisela Viridiana Villafañe-Barajas, Abigail González-Valdez, Veronica Roxana Flores-Vega, Roberto Rosales-Reyes, Marco Elias Gudiño-Zayas, Thelma Rizo-Pica, Adelfo Escalante, Miguel Cocotl-Yanez.

**Funding acquisition:** Gloria Soberón-Chávez, Miguel Cocotl-Yanez.

**Investigation:** Misael Josafat Fabian Del-Olmo, Luis Fernando Montelongo-Martínez, Gisela Viridiana Villafañe-Barajas, Veronica Roxana Flores-Vega, Roberto Rosales-Reyes, Marco Elias Gudiño-Zayas, Thelma Rizo-Pica, Raunel Tinoco-Valencia, Adelfo Escalante, Gloria Soberón-Chávez.

**Methodology:** Misael Josafat Fabian Del-Olmo, Luis Fernando Montelongo-Martínez, Gisela Viridiana Villafañe-Barajas, Abigail González-Valdez, Veronica Roxana Flores-Vega, Roberto Rosales-Reyes, Marco Elias Gudiño-Zayas, Thelma Rizo-Pica, Raunel Tinoco-Valencia.

**Project administration:** Miguel Cocotl-Yanez.

**Resources:** Gloria Soberón-Chávez, Miguel Cocotl-Yanez.

**Supervision:** Miguel Cocotl-Yanez.

**Validation:** Misael Josafat Fabian Del-Olmo, Roberto Rosales-Reyes, Marco Elias Gudiño-Zayas, Raunel Tinoco-Valencia, Gloria Soberón-Chávez, Miguel Cocotl-Yanez.

**Writing – original draft:** Misael Josafat Fabian Del-Olmo, Luis Fernando Montelongo-Martínez, Miguel Cocotl-Yanez.

**Writing – review & editing:** Misael Josafat Fabian Del-Olmo, Luis Fernando Montelongo-Martínez, Gisela Viridiana Villafañe-Barajas, Abigail González-Valdez, Roberto Rosales-Reyes, Raunel Tinoco-Valencia, Adelfo Escalante, Gloria Soberón-Chávez, Miguel Cocotl-Yanez.

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
