## [Decision Letter · Decision Letter 0]

30 Sep 2025

Dear Dr. Cocotl-Yanez,

Thank you for submitting your manuscript to PLOS ONE. After careful consideration, we feel that it has merit but does not fully meet PLOS ONE’s publication criteria as it currently stands. Therefore, we invite you to submit a revised version of the manuscript that addresses the points raised during the review process.

We look forward to receiving your revised manuscript.

Kind regards,

Rajesh P. Shastry, Ph.D

Academic Editor

PLOS ONE

Journal Requirements:

“MC-Y research was supported by Programa de Apoyo a Proyectos de Investigación e Innovación Tecnológica (PAPIIT) DGAPA, Universidad Nacional Autónoma de México (UNAM), grant number IA200823. GS-Ch research was supported in part by DGAPA, PAPIIT UNAM grant IN202325 and by SECIHTI grant CBF-2025-I28. The funders had no role in the study design, data collection and analysis, decision to publish, or preparation of the manuscript.”

“MJF-O, LFM-M and GVV-B are students of the Programa de Maestría y Doctorado en Ciencias Bioquímicas at the Universidad Nacional Autónoma de México (UNAM), and VRF-V is a student of the Departamento de Biomedicina Molecular at CINVESTAV-IPN. They received a fellowship from SECIHTI (CVU 1230606, 927093, 2055397, and 1165608 respectively). MC-Y research was supported by Programa de Apoyo a Proyectos de Investigación e Innovación Tecnológica (PAPIIT) DGAPA, Universidad Nacional Autónoma de México (UNAM), grant number IA200823. GS-Ch research was supported in part by DGAPA, PAPIIT UNAM grant IN202325 and by SECIHTI grant CBF-2025-I28. The authors would like to thank Dr. Brenda Rosario Sandoval Meza, part of the Medical Translation Area at the Research Division, Faculty of Medicine, Universidad Nacional Autónoma de México, for proofreading this paper. “

“MC-Y research was supported by Programa de Apoyo a Proyectos de Investigación e Innovación Tecnológica (PAPIIT) DGAPA, Universidad Nacional Autónoma de México (UNAM), grant number IA200823. GS-Ch research was supported in part by DGAPA, PAPIIT UNAM grant IN202325 and by SECIHTI grant CBF-2025-I28. The funders had no role in the study design, data collection and analysis, decision to publish, or preparation of the manuscript.”

Reviewers' comments:

Reviewer's Responses to Questions

**Comments to the Author**

1. Is the manuscript technically sound, and do the data support the conclusions?

Reviewer #1: Yes

Reviewer #2: Yes

2. Has the statistical analysis been performed appropriately and rigorously?

Reviewer #1: Yes

Reviewer #2: I Don't Know

3. Have the authors made all data underlying the findings in their manuscript fully available?

Reviewer #1: Yes

Reviewer #2: Yes

4. Is the manuscript presented in an intelligible fashion and written in standard English?

Reviewer #1: Yes

Reviewer #2: Yes

Reviewer #1: This manuscript presents evidence that an RsmA-deficiency mutant ID4365 of Pseudomonas aeruginosa is suitable for producing high amount of pyocyanin. The experiments provided are necessary and the results are reasonable and logical. English is very perfect. Only on problem should be clarified: pyocyanin is reported as one of virulent factors in many published works. The mutan P. aeruginosa IDrsmA produces much more pyocyanin in some medium and under specific optimal culture conditions.

Did you check whether the pyocyanin serves as virulence factor biosynthesized and released in other cells or animals (not Galleria mellonella and mice). This problem should be clarified and discussed in discussion section at least.

Reviewer #2: Dear author,

this is a good article that describes the production of substances of biotechnological interest in an avirulent strain, thereby simplifying the safety processes involved in obtaining these compounds. The manuscript is very well written and is relevant in the scientific field.

**Do you want your identity to be public for this peer review?** For information about this choice, including consent withdrawal, please see our Privacy Policy

Reviewer #1: **Yes: ** Yihe Ge

Reviewer #2: **Yes: ** Rosa del Campo

---

## [Author Response · Author response to Decision Letter 1]

8 Oct 2025

Reviewer #1: This manuscript presents evidence that an RsmA-deficiency mutant ID4365 of Pseudomonas aeruginosa is suitable for producing high amount of pyocyanin. The experiments provided are necessary and the results are reasonable and logical. English is very perfect. Only on problem should be clarified: pyocyanin is reported as one of virulent factors in many published works. The mutant P. aeruginosa IDrsmA produces much more pyocyanin in some medium and under specific optimal culture conditions.

Did you check whether the pyocyanin serves as virulence factor biosynthesized and released in other cells or animals (not Galleria mellonella and mice). This problem should be clarified and discussed in discussion section at least.

R= Thank you very much for your comment. We did not check it. However, considering the characteristics of the IDrsmA strain related to its virulence factors (such as lower levels of elastase and rhamnolipids, along with an inactivated T3SS), similar results could be obtained using additional animal or cell models. This is discussed in the discussion section, lines 477-482.

Reviewer #2: Dear author,

this is a good article that describes the production of substances of biotechnological interest in an avirulent strain, thereby simplifying the safety processes involved in obtaining these compounds. The manuscript is very well written and is relevant in the scientific field.

R= Thank you very much.

---

## [Decision Letter · Decision Letter 1]

4 Nov 2025

The rsmA mutant from Pseudomonas aeruginosa ID4365 is a non-virulent strain that is suitable for pyocyanin and phenazine-1-carboxylic acid production.

PONE-D-25-46541R1

Dear Dr. Cocotl-Yanez,

We’re pleased to inform you that your manuscript has been judged scientifically suitable for publication and will be formally accepted for publication once it meets all outstanding technical requirements.

Kind regards,

Rajesh P. Shastry, Ph.D

Academic Editor

PLOS ONE

Additional Editor Comments (optional):

Reviewers' comments:

Reviewer's Responses to Questions

**Comments to the Author**

Reviewer #3: All comments have been addressed

2. Is the manuscript technically sound, and do the data support the conclusions?

Reviewer #3: Yes

3. Has the statistical analysis been performed appropriately and rigorously?

Reviewer #3: Yes

4. Have the authors made all data underlying the findings in their manuscript fully available?

Reviewer #3: Yes

5. Is the manuscript presented in an intelligible fashion and written in standard English?

Reviewer #3: Yes

Reviewer #3: This is a very interesting study regarding phenazines and their relations with Pseudomonas aeruginosa strains. Further research in this field is needed, in order to enhance their medical and biotechnological role.

**Do you want your identity to be public for this peer review?** For information about this choice, including consent withdrawal, please see our Privacy Policy

Reviewer #3: No

---

## [Editor Report · Acceptance letter]

PONE-D-25-46541R1

PLOS ONE

Dear Dr. Cocotl-Yanez,

I'm pleased to inform you that your manuscript has been deemed suitable for publication in PLOS ONE. Congratulations! Your manuscript is now being handed over to our production team.

Kind regards,

on behalf of

Dr. Rajesh P. Shastry

Academic Editor

PLOS ONE